# Untangling Urban Sprawl and Climate Change: A Review of the Literature on Physical Planning and Transportation Drivers

**Qiu Feng \* and Pierre Gauthier**

Department of Geography, Planning and Environment, Concordia University, 1455 Boulevard de Maisonneuve O, Montréal, QC H3G 1M8, Canada; pierre.gauthier@concordia.ca
\* Correspondence: qiu.feng@mail.concordia.ca

**Abstract:** Significant efforts have been dedicated to studying the linkages between urban form, fossil energy consumption, and climate change. The theme of urban sprawl helped to federate a significant portion of such efforts. Yet, the research appears fragmented, at stems from different disciplines and mobilizes different methods to probe different aspects of the issue. This paper seeks to better understand the status of knowledge concerning the linkages between sprawl and climate change through a critical review of the literature published between 1979 and 2018. The exercise entailed revisiting how sprawl has been defined, characterized and measured, and how such parameters have informed the research themes and the approaches mobilized to study its impacts on climate change. For, sprawled environments contribute the climate change directly and indirectly, due to the individual or combined effects of its land use, land cover, urban form, and transportation characteristics. The results indicate that sprawl's impacts have been mainly investigated in three principal streams of research and based on a limited number of factors or combinations of factors. Though a strong consensus emerges on the negative environmental costs of sprawl, including toward climate change, there remain ambiguities when trying to untangle and weigh specific causes.

**Keywords:** climate change; urban sprawl; urban form; land use; physical planning; urban transportation; greenhouse gas emission; energy consumption

## 1. Introduction

This research aimed at answering a pretty straightforward question: "what does the research say about the links between urban sprawl and climate change." The exercise stemmed from the desire of the two largest regional planning bodies of the Province of Québec to get a synthetic summary of the research approaches and key findings on the matter, in the context of the revision of their Regional Plans. The general consensus in the applied planning literature is that sprawled environments epitomized, in a North American context particularly, by post-Second World War automobile-based suburban development models, are the cause of environmental degradations while contributing to climate change. There are plenty of empirical pieces of evidence to support such tenets. Yet, upon closer examination, the issue appears more elusive than it might look at first sight. Establishing the linkages between sprawl and climate change is complicated by many factors, starting with the inherent complexities of the interactions between the natural and built environments and the difficulties to untangle the direct, indirect, mutual impacts between them, on the one hand, and difficulty of tackling the sprawl phenomenon itself, on the other hand.

The determination of the impacts of urban sprawl on climate change and their measurement can, and do, vary significantly depending on how sprawl itself is conceptualized and measured. Though the overall negative environmental impacts of sprawl are well established, seemingly inconclusive or contradictory results on their precise causes are not rare. Some of these conundrums are impeding our ability to fully understand the

connections between sprawl and climate change and to articulate a proper public policy response. This paper presents key conclusions stemming from a survey of pertinent literature probing the links between urban sprawl and climate change. It sets about, firstly, to bring some conceptual clarity to the notion of sprawl, while charting, secondly, the scientific production that addresses direct and indirect links between the latter and climate change. Those links are examined through the lenses of energy consumption, and greenhouse gas (GHG), and carbon dioxide ($CO_2$) emissions. While tremendous efforts have already been made in exploring the relationships between urban forms, including sprawled environments, energy consumption, and climate change [1–6], it is very difficult for specialists and interested stakeholders to build a synthetic picture and to make sense of seemingly disparate research.

Following a brief discussion on the approach mobilized to review the pertinent literature and gather pertinent materials, the following sections introduce sprawl's definitions, characterizations, and quantification methods. A conceptual diagram has proposed that charts what we deem the physical planning drivers of climate change as well as their direct, indirect, and combined impacts. Key research approaches and findings on the links between sprawl and climate change are then presented according to three main streams revealed by the review of the pertinent literature. A discussion follows on the complementarities and limitations of the research programs while pointing to some ambiguities and apparent inconsistencies.

## 2. Materials and Methods

A two-pronged approach was adopted for identifying and analyzing the pertinent literature probing the linkages between urban sprawl and climate change. The first step centered on 7 literature reviews published between 1994 to 2015 (since expanded to include two reviews published in 2020 [7,8]) (see Table 1) on urban sprawl per se, or the environmental impacts of sprawl. The works and research themes covered in these 7 reviews laid a solid foundation for further search, by allowing, firstly, to review definitions, conceptualizations, and characterizations of sprawl, and secondly, to identify potent keywords and combinations of keywords that address the linkages between sprawl, or key aspects of it, and climate change. The second step entailed conducting a search relying on ISI's Web of Science®® database from 1979 until 2018. Various iterations of (urban sprawl OR sprawl) AND (climate change OR global warming OR greenhouse gas emission* OR $CO_2$) AND (transportation OR land use* OR land cover change) were performed. A preliminary assessment, including the probing of the articles' bibliographies, was followed by a more thorough examination. The analytical approach mobilized is the critical literature review. This approach does not seek exhaustivity. It is meant to document, compare and contrast contributions from different theoretical, methodological, and epistemological perspectives [9]. The exercise prioritized quantitative studies concerned with the material and spatial manifestations of sprawl in relation to outputs that contribute to climate change while excluding studies centered on economic, social, cultural, or technological factors. North American contexts have been given more weight compared to other settings. A satisfying level of saturation was reached after selecting and perusing some 220 academic contributions, including 9 literature reviews, in the fields of (by decreasing order of importance): environmental sciences, environmental studies, urban studies, ecology, water resource management, sustainable technologies, multidisciplinary geoscience, geography, and planning. Those contributions constitute the core material of this review.

**Table 1.** Reviews of either urban sprawl, the environmental impacts of urban sprawl, or both.

| Review Author (s), Year of Publication Title of the Review | Significance of the Study |
| --- | --- |
| Ismael (2020) <br> *Urban form study: the sprawling city—review of methods of studying urban sprawl* | selectively reviewed important existing and novel methods to study and measure urban sprawl from the field of urban geography. |
| Rubiera-Morollon and Garrido-Yserte (2020) <br> *Recent Literature about Urban Sprawl: A Renewed Relevance of the Phenomenon from the Perspective of Environmental Sustainability* | reviewed the literature on sprawl since 2000, mainly from 2010–2020, while identifying key factors behind its renewed relevance with respect to environmental sustainability in relation to new methodological and recent theoretical advances. |
| Ewing and Hamidi (2015) <br> *Compactness versus Sprawl: A Review of Recent Evidence from the United States* | revisited the debates about urban sprawl and compact city and summarized the pertinent literature on characteristics, measurements, causes, impacts, and remedies of sprawl. |
| Yigitcanlar and Kamruzzaman (2014) <br> *Investigating the interplay between transport, land use and the environment: a review of the literature* | surveyed publications from database-ScienceDirect from 1990 and onwards on the latest empirical approaches and best practices worldwide to examine the interplay between transport, land use, and the environment. |
| Wilson and Chakraborty (2013) <br> *The Environmental Impacts of Sprawl: Emergent Themes from the Past Decade of Planning Research* | extended and updated Johnson's (2001) work by collecting articles published since 2001 related to the environmental impacts of sprawl. |
| Burchell et al. (2002) <br> *Costs of Sprawl, 2000.* | analyzed urban sprawl, its impacts on resources, personal costs of sprawl, benefits of sprawl, and ways to reduce its negative effects. |
| Johnson (2001) <br> *Environmental impacts of urban sprawl: a survey of the literature and proposed research agenda* | one of the most widely cited and influential reviews associated with the environmental impacts of sprawl. |
| Burchell et al. (1998) <br> *The Costs of Sprawl—Revisited* | provided "a detailed examination of most of the information that can be assembled on both sprawl and its costs . . . " (p.ii) |
| Ewing (1994) <br> *Characteristics, Causes, and Effects of Sprawl: A Literature Review* | reviewed literature on definitions, characteristics, and effects of urban sprawl. |

## 3. Results

The current cycle of climate change is attributable to natural (volcanic activity and solar output for the most part) and anthropogenic drivers. There is an overwhelming consensus to the effect that the GHG emissions caused by human activities are the most important cause of climate change [6,10–14]. Among all of the GHGs, carbon dioxide ($CO_2$) is the most detrimental contributor to global climate change [6,15,16]. The 2007 IPCC report identifies two primary anthropogenic drivers of increases in atmospheric $CO_2$: fossil fuel combustion and land-use change. Between 1970 and 2010, around 78% of the total GHG emissions increase was caused by fossil fuel combustion and industrial processes [17].

Cities are already responsible for approximately 80% of the overall GHG/$CO_2$ emissions [18], while urbanized populations are expected to double, as rural populations level off or decline [19] (p.3133). The Contribution of Working Group III to the Fifth Assessment Report of the Intergovernmental Panel on Climate Change identifies urban form as a significant driver for GHG emissions [17]. The term urban form designates the spatial arrangement of buildings and infrastructures in urbanized contexts.

The GHG/$CO_2$ emissions pertaining to urban form are predominantly attributable to the fossil energy consumption linked to transportation dynamics and for the heating and cooling in buildings [17,20,21]. Yigitcanlar and Kamruzzaman noted that "transport and land uses are the two major sectors that contribute most in emitting $CO_2$ in the environment" [15] (p.2121). As will be discussed, built forms, land use, land cover, and transportation respectively contribute to the GHG/$CO_2$ emissions on their own, while impacting one another. The following paragraphs will highlight some of those key dynamics.

*3.1. Defining, Characterizing, and Measuring Sprawl*

3.1.1. Defining Sprawl

The forms assumed by urbanization have evidently evolved in the course of history. For thousands of years, the forms of cities had been essentially predicated on walking. Such conditions have informed the architectural and urban configurations as well as the spatial distribution of amenities and activities. Some ancient cities have housed large populations. Thebes in Egypt reached the milestone of 100,000 inhabitants around 1000 years BCE, while Rome, Beijing, and London reached the million mark around 100 CE for the first and the turn of 19th century for the latter ones [22]. Yet, even the most populous cities displayed a spatial extension limited to a radius of 5 km or so, corresponding to an hour of walking from the periphery to the center. Such cities had to "grow from within," which entailed increased densities of populations, buildings, and activities. The introduction of trains and streetcars triggered spatial expansion and the advent of new urban configurations between 1850 to 1950. However, it's the introduction of the automobile in the 1930s, and the generalization of automobility after the Second World War, that would enable urban sprawl, i.e., the spatial dispersion of populations and activities on large territories. Sprawl has become the predominant contemporary urban form at great environmental costs. It is now seen as a "fundamental cause of unsustainability in cities" [23] (p.64).

Urban sprawl has been discussed and researched from a wide diversity of perspectives in the urban studies and planning literature. Once described as an "American zeitgeist" [24], sprawl is becoming a global phenomenon. Sprawl has been substantially altering the physical and spatial structures as well as the functioning of cities where it prevails, though the rates and patterns of sprawl vary in different parts of the world. Such variability is one of the factors that explain the difficulty to characterize the phenomenon. There is no unitary or consensual definition of sprawl in the literature, as regularly stressed by some of its most dedicated observers [25–32]. Various terms such as dispersion, suburban sprawl [33], suburbanization, suburbia and edgeless city [34] have been used in trying to qualify, contextualize or better denote sprawl, but none took hold as an appropriate alternative. The definitions of sprawl are strongly informed by the disciplinary, theoretical, and epistemological perspectives of the definers [35] (p. 3). Moreover, the term alternatively refers to a process, an outcome, or to specific material and spatial manifestations. Adding to the difficulty, each of those instances is responsive to their context and is subject to geographical and temporal variability. A collection of definitions, including from some of the most recognized experts in the field illustrate the last point (Table 2) while highlighting the difficulty to capture the complexity and multidimensional reality of sprawl in a single definition.

**Table 2.** The definitions of sprawl: summary of findings [36] (p. 13–14).

| Authors/ Year of Publication | Definition of Urban Sprawl | Particularity of the Definition |
|---|---|---|
| Burchell et al., (1998) | "Sprawl refers to a particular type of suburban peripheral growth." (p. 6). | They stress that sprawl's distinguishing trait: density, should be assessed in relative terms: i.e., especially density should be set "in context", relative to localized circumstances (cultural, geographical, etc.) and relative to the sound use of the resources in that particular context. |
| Sierra Club, (1998) | "low-density, automobile-dependent development beyond the edge of service and employment areas". | The definition stresses some of the sprawl's spatial characteristics (density, position relative to service, etc.) and effects (automobile dependence). |

| Authors/ Year of Publication | Definition of Urban Sprawl | Particularity of the Definition |
|---|---|---|
| Nelson and Duncan, (1999) | "Unplanned, uncontrolled, and uncoordinated single-use development that does not provide for an attractive and functional mix of uses and/or is not functionally related to surrounding land uses and which variously appears as low density, ribbon or strip, scattered, leapfrog, or isolated development." (p. 1). | The definition mixes normative and affective criteria (functional, attractive), spatial attributes (scattered, isolated, etc.), and the characterization of development processes (uncontrolled, etc.). |
| Barners et al., (2001) | "sprawl as a pattern of land-use/land cover conversion in which the growth rate of urbanized land (land rendered impervious by development) significantly exceeds the rate of population growth over a specified time period, with a dominance of low-density impervious surfaces." (p. 4). | The definition refers to urbanization processes (land cover conversion, the ratio of land urbanized/population growth) and the resulting spatial patterns (land-use) and spatial properties (density, impervious surfaces, etc.). |
| Gaslter et al., (2001) | "Sprawl (n.) is a pattern of land use in a UA that exhibits low levels of some combination of eight distinct dimensions: density, continuity, concentration, clustering, centrality, nuclearity, mixed uses, and proximity." (p. 685). | Sprawl is defined in purely spatial terms, as the pattern resulting from the combination of eight properties manifested at "low-levels" of intensity. The said properties allow quantification, hence inaugurating the "first multidimensional measures of sprawl by disaggregated land-use patterns into eight different dimensions" (Ewing and Hamidi, 2014). |
| Jaeger et al., (2010) | "A landscape suffers from urban sprawl if it is permeated by urban development or solitary buildings." (p. 400). | Sprawl is defined in spatial and topological terms and as a gradient, which takes into consideration the developed, or "built" land cover. |
| Jaeger and Schwick, (2014) | "A landscape suffers from urban sprawl if it is permeated by urban development or solitary buildings and when land uptake per inhabitant or job is high". (p. 296). | Updated Jaeger et al. 2010 definition, sprawl is defined in spatial and topological terms and as a gradient, which takes into consideration the developed, or "built" land cover as well as land uptake (expressed in ratios inhabitants/land area and jobs/land area). |
| Ewing, Tian, and Lyons, (2018) | "sprawl is operationally defined as low density, single-use, uncentered, or poorly connected development". (p. 96). | This operational definition of sprawl centers on four spatial characters affecting the distribution of people and urban functions (land-use) and the configurational properties of the street network (connectivity). |

While there is no unitary definition of sprawl in the literature, its definitions and characterizations revolve around three highly recurring aspects or characters. Sprawl manifests patterns of land development marked by low-intensity (expressed in densities of population and activities) and spatially segregated land uses (comprised of residential and other functions such as commercial, leisure, and economic production). Such spatial patterns are enabled by, and heavily dependent on, automobility. Sprawl is often contrasted with and compared (in qualitative and quantitative terms) to its polar opposite, deemed the "compact city." The latter typically refers to environments urbanized prior to the generalization of the automobile that are characterized by high densities and diversities of land-uses, and that are amenable to walking and to the deployment and use of public transit [2,6,37]. Though evocative, the compact city notion remains elusive and difficult to

operationalize (and to measure, for instance) for the same reasons that affect the notion of sprawl. The difficulty stems from the relativity of the conditions that are described or probed. Whether an environment can be deemed compact or sprawled is relative. There is no way to establish criteria and thresholds empirically, let alone universal ones. At best, sprawl and compactness describe conditions at the opposite ends of the development on a continuum [2,32]. Such categorization overlooks the intermediary conditions and turns a blind eye to atypical combinatory patterns.

Feng and Gauthier propose a definition of sprawl that accounts for the context: " . . . the term sprawl denotes an urbanization process that produces low-intensity modes of occupation of the land. [Sprawl] is characterized by built and spatial forms that are suboptimal in serving their purposes when taking into consideration their geographical, cultural, and technological contexts and local historical precedents" [36] (p. 16). They posit further, referring to the three sustainable development pillars, that sprawl "produces a suboptimal return on investment, environmentally, socially, and economically speaking, for the community" [36] (p. 16). Feng and Gauthier's conceptualization does not preclude the possibility of measuring sprawl or some of its key components but stresses that the results need to be interpreted relative to the context [36]. While sprawl's environmental impacts can be measured in relative terms, against normative sustainable criteria and benchmarks, or in absolute terms, based on concrete GHG emissions outputs for instance. The difficulty in the latter case lays in the ability to measure accurately sprawl itself and to untangle the causes from the effects of the intertwined characteristics of the phenomenon as we will see.

### 3.1.2. Measuring Sprawl Key Characteristics

Sprawl is a multi-dimensional phenomenon that requires different measures for each dimension [30,38]. Pendall stated that "the measurement of sprawl is not straightforward, partly because of the variation in how sprawl is defined" [39] (p. 558). Various approaches and methods have been developed to quantify sprawl. Ewing and Hamidi classified the efforts to quantify the extent of sprawl into three stages [2]. The early research, prior to the year 2000, was crude and unidimensional, exclusively or merely focusing on density; the 2001–2010 period has featured multi-level, multi-dimensional, and multi-disciplinary approaches; the subsequent stage from 2011 aimed also to tackle changes or trends in the degrees of urban sprawl [2]. Each method has its own advantages and limitations. Some studies using different methods have delivered inconsistent, and seemingly divergent or contradictory results. As always, caution is needed when interpreting results. Sprawl is characterized by intertwined sets of characters that interact with and influence each other. Its measurement is also particularly sensitive to spatial resolution and the modifiable areal unit problem (i.e., the variability of the shape and scale of the spatial unit against which the data is aggregated) [40,41]. In recent years, major advances in geospatial technologies, including GIS, remote sensing, and photogrammetric techniques have expanded the researchers' toolbox [7,8]. They have allowed in particular to measure sprawl physical characteristics more accurately across a variety of international contexts while facilitating comparative analysis [8].

The most common categories of variables mobilized to measure properties associated with the three main characters of sprawl have been identified, by relying on inductive and deductive reasoning while probing the literature on sprawl characterization and measurement. Table 3 summarizes these findings.

**Table 3.** Sets of variables are used to measure sprawl.

| Category | Character | Variable Name | Definition |
|---|---|---|---|
| Urban sprawl | Density | Population density | Density is most commonly defined as population/housing or employment density, which are measured per unit of analysis. |
| | | Residential density | |
| | Land use | D variables, first 3 Ds the 5Ds, density, diversity, and design, destination accessibility and distance to transit) | The number of different land use in a given area (at a mesoscale: neighborhood or activity center, land use patterns are characterized by various measures of land use mix within neighborhoods and activity centers). Two land-use mix measures have become most accepted: an entropy index and a dissimilarity index. |
| | Transportation/automobile dependence | Commute time | Vehicle hours traveled |
| | | Trip distance: VMT | Vehicle miles traveled (or vehicle hours traveled) "is a primary performance indicator for land use and transportation" (Ewing et al., 2014, p. 3080). |
| | | Mode split | Probability (or percentage) of commuting by automobile, transit (rail or bus), or by non-motorized mode (walking/cycling); others also include moped, motorcycle, taxi. |

Density has been used as one of the chief measurements or the sole indicator of urban sprawl in many early studies. Expressed by a number of people/housing units or jobs per geographical area unit (acre, hectare, and $km^2$ for instance), it is the most commonly used measurement of sprawl in the literature. Technically, density variables measure the intensity of a specific land-use category in an area of reference. They are a sub-set of land-use variables. Land use variables are routinely referred to as the so-called "D variables." They are centered on the compositional and configurational characteristics of the land allocation in a geographical area of reference, as well as on accessibility to specific amenities (jobs, transit stops) or overall accessibility by foot (relying on topological variables as proxies). Cervero and Kochelman coined the original expression "three Ds," which stands for: density, diversity (land-use composition), and design (accessibility based on a place's spatial characteristics) [42], later expanded to include destination accessibility and distance to transit, referred to as the fourth and fifth D variables [43] (p. 200). The land-use variables are typically measured at the neighborhood scale by using census data, agency data, or data that can be derived from GIS. Sprawl is almost indissociable from heavy reliance on automobiles. In the sprawling literature, auto dependence has been measured using proxies such as the modal share, and the total amount of time or the distance traveled (typically by car, expressed in vehicle miles/kms traveled—VMT/VKT). The most widely used and primary performance indicator in land use and transportation studies relies on VMT/VKT measurements [44] (p. 3079). Not surprisingly, the methods developed to analyze the main characters of urban sprawl separately or in combination, have informed the research on sprawl and climate change. Another important facet has been the analysis of the land cover, which refers to the composition of the ground surface itself.

Figure 1 illustrates the links between urban sprawl and climate change. It charts, more specifically, what we deem the physical planning (land use and land cover) and transportation drivers of change; the elements analyzed and the type of measurements; the dynamics involved; their effects and consequences, and finally the outcomes (i.e., contribution to climate change or vulnerabilities with respect to the impacts of climate change).

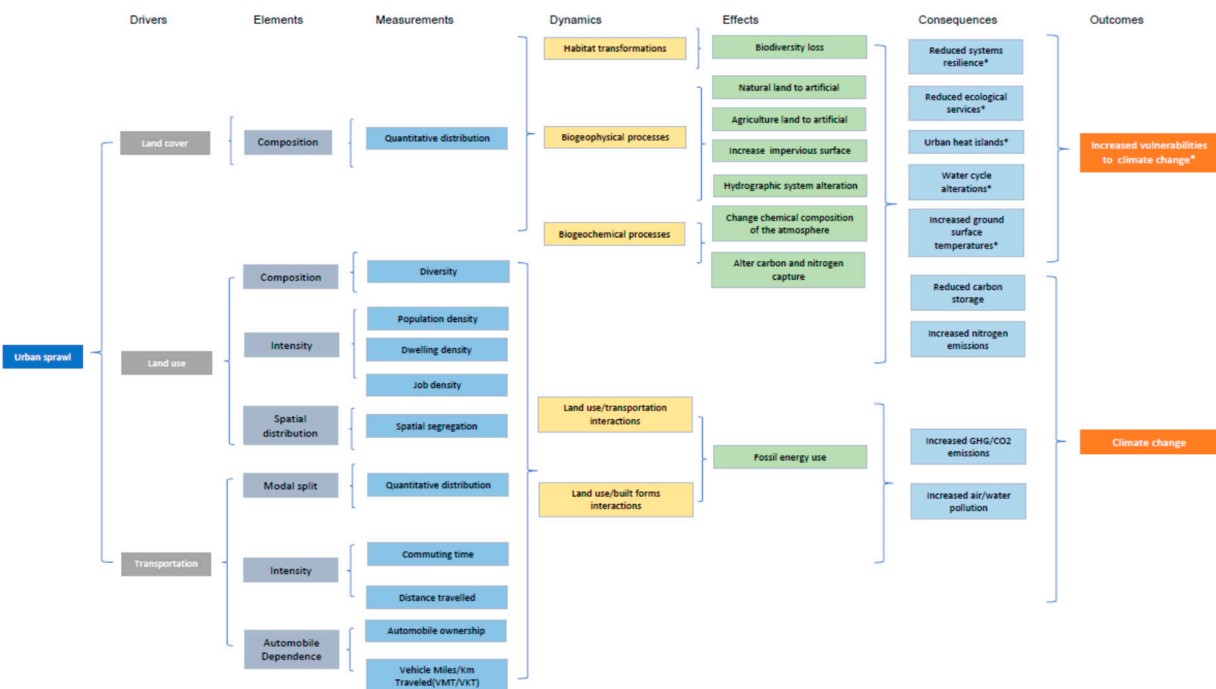

**Figure 1.** Conceptual diagram of the relationship between urban sprawl and climate change.

### 3.2. Sprawl and Climate Change

The following sections report on the three most important "streams of research" investigating the relationships between sprawl and climate change as revealed by the literature review, while summarizing some of their key conclusions. The first stream focuses on the linkages between density, fossil energy consumption, and GHG/$CO_2$ emissions. The second one is considering the dynamics between land use energy consumption and emissions, and between land cover change and the carbon balance. The third stream is centered on the impacts of transportation dynamics on climate change. The joined impacts of land use and transportation interactions and land use and built forms on fossil energy use and GHG/$CO_2$ emissions are also addressed.

3.2.1. The Impacts of Urban Density on Fossil Energy Consumption and GHG/$CO_2$ Emissions

Density was and still is one of the most widely used indicators to measure land-use and urban form intensity. "Density, or more specifically, low density, is one of the cardinal characteristics of sprawl" [24] (p. 6). Unsurprisingly, early research efforts to quantify the impacts of urban form on energy consumption and GHG/$CO_2$ emissions centered on density. Different measures of density, such as the number of inhabitants or dwellings per unit of space have been used as proxies for land-use intensity or urban form compactness. The most important line of inquiry focuses on urban density's links with fossil energy consumption and GHG/$CO_2$ emissions due to transportation. The second line of inquiry centers on the energy consumption and emissions of the buildings themselves (heating, cooling, etc.) depending on the density.

Early attempts to explore the links between population density on energy consumption and GHG/$CO_2$ emissions, which focused on the outputs of the transportation patterns associated with various land-use and urban form configurations have produced sound and valuable insights into nature and the impacts of these intertwined relationships [20,45–47]. The seminal study by Newman and Kenworthy of 32 cities from around the world shows a clear negative relationship between population density and fuel consumption [20]. Subsequent efforts have produced a deeper understanding by probing the trends and patterns of changes in the said relationships over a long period [48].

Low-density residential environments composed of single-family houses are a defining characteristic of sprawl. Many scholars investigate the relationship between residential density, the level of GHG emissions related to both transportation patterns and building energy use [49–52]. Different residential densities are typically represented by different neighborhood types—Generally classified as either traditional (from the pre-automobile era) or suburban. Living and transportation arrangements in low-density suburban contexts are compared with urban center's living, characterized by high-density apartment buildings, to measure their respective energy consumption and levels of GHG emissions. However, as Osorio et al. argued, probing the energy consumption of buildings is complicated, and even more so at the neighborhood or network of buildings scales [53]. Low-density, detached single-family buildings and urban fabrics are associated with higher energy consumption and GHG/$CO_2$ emissions, but specific contributing factors are difficult to untangle and measure precisely.

In general, the literature produces very strong pieces of evidence showing that population/residential densities are negatively correlated with energy consumption and GHG/$CO_2$ emissions and that increasing density results in lower such outputs.

Table 4 summarizes, in chronological order, the approach, context, and key findings of selected research investigating the relationships between density, energy consumption, and GHG emissions.

**Table 4.** The impacts of urban density on fossil energy consumption and GHG/$CO_2$ emissions [36] (p. 39–40).

| Author(s) Year of Publication | Type of Density | Relationship Studied with Density | Geographical Context | Main Results |
|---|---|---|---|---|
| Newman and Kenworthy (1989) | Population density | Gasoline consumption per capita | 32 global cities | Per capita gasoline consumption is negatively correlated with population density. |
| Norman, MacLean, and Kenned (2006) | Residential density | Energy use and GHG emissions | Toronto | $CO_2$ equivalent emissions are 60% less for high-density than for low-density development. |
| Nelson and Duncan (1999) | Residential building density | GHG emissions | Toronto | Top ten in terms of GHG emission were all located in the low-density tracts. |
| Andrews (2008) | Urban density | GHG emissions distribution along the rural-to-urban gradient | United States Canadian cities | Per-capita $CO_2$ emissions vary widely following an inverted "U" shape, with post-war suburbs at the pinnacle. |
| Ewing and Rong (2008) | House size and type | Housing types and energy consumption | United States | Houses located in compact counties require roughly 20% less primary energy than those in sprawling counties. |

**Table 4.** *Cont.*

| Author(s) Year of Publication | Type of Density | Relationship Studied with Density | Geographical Context | Main Results |
|---|---|---|---|---|
| Taniguchi, Matsunaka, and Nakamichi (2008) | Population density | Per capita automobile $CO_2$ emissions | 38 Japanese cities | Density negatively correlated with automobile $CO_2$ emissions Per-capita automobile $CO_2$ emissions increased in all city types between 1987 and 2005. |
| Glaeser and Kahn (2010) | Population density | Household emissions | 66 major US cities | Gasoline usage is negatively correlated with population density and positively correlated with distance from downtown. |
| Kim and Brownstone (2010) | Residential density | Household annual mileage traveled and fuel consumption | United States | Households residing in an area that is 1000 housing units per square mile denser drive 1500 (7.8%) fewer miles per year and consume 70 (7.5%) fewer gallons of fuel than households in the less dense areas. |
| Ala-Mantila, Junnila, and Heinonen (2013) | Residential types (Semi-detached and detached houses, apartment buildings) | Consumption-based carbon footprints by residential types | Finland | Low-rise lifestyle causes approximately 26% more emissions than high-rise. |
| Pitt (2013) | Residential types (attached, multifamily, single-family detached housing) | Residential GHG emissions and energy consumption for future housing development | United States | On average, attached homes and multi-family structures are more energy-efficient than single-family detached housing types. |
| Ala-Mantila, Heinonen, and Junnila (2014) | Housing and household types | Consumption-based carbon footprints by housing and household types | Finland | Rural lifestyle related to the highest GHG emissions. Emissions decrease as density increases while moving towards city centers. |
| Fercovic and Gulati (2016) | Population density | Average household emissions | Canadian cities | Denser cities produce fewer emissions than low-density ones. Average household emissions across all cities over time are falling. |
| Estiri (2016) | Households housing arrangement (city and suburban) | Energy consumption by households | United States | On average, US suburban households consume more energy in residential buildings than their city-dweller counterparts. |

### 3.2.2. The Impacts of Land Use and Land Cover on Fossil Energy Consumption and GHG/CO$_2$ Emissions

The term land use refers to the composition and configurations of the land surface utilization (for housing, work, leisure, transportation, etc.). Land cover denotes the nature and the composition of the surface on the ground (such as forest land, grassland, wetlands, anthropogenic biomes of crops, or artificial infrastructures and buildings). The research probing the links between land use patterns and urban development dynamics in relation to energy consumption and GHG/CO$_2$ emissions touches: 1. on direct outputs and direct impacts on the carbon balance; 2. on indirect outputs linked to transportation dynamics associated with land use patterns and; 3. on increased environmental vulnerabilities.

Sprawl entails massive land cover changes involving the artificialization of natural or cultivated land. Such transformations translate into deforestation and grasslands losses; loss of valuable arable land; the creation of extensive impervious surfaces and; extensive construction of buildings and roads. Land conversion causes significant losses of biomass while altering natural habitats and ecosystems. Land cover change associated with sprawl contributes to climate change by reducing the carbon capture and storage capacities [54]. It furthers the ecosystem's vulnerabilities stemming from bioclimatic transformations induced by climate change itself and compromises these ecosystems' ability to mitigate the impacts of extreme weather events such as heavy rains [3,12].

The research investigating the relations between the composition and configurations of land utilization, energy consumption, and GHG/CO$_2$ emissions, usually focuses on the transportation implications of land use conditions in sprawled and compact environments respectively. The composition and spatial distributions of urban activities and functions influence people's travel behavior by "affecting decisions about how much, where, when, and how to get around" [55] (p. 2). The degree of land use mix not only correlates with VMT/VKT but exerts also an influence on the choice of the mode of transportation.

A majority of empirical studies surveyed herein conclude that more mixed land uses and compact urban forms that are complemented by a good public transit system and a well-connected and easily accessible street network are associated with fewer VMT/VKT, lower levels of GHG/CO$_2$ emissions, and energy consumption, as well as a lesser dependence to the automobile when compared to sprawled contexts [6,56–58].

Table 5 summarizes, in chronological order, the contexts, methods, and main findings of studies probing the impacts on energy consumption and GHG/CO$_2$ emissions of land use/land cover change, composition, and configurations.

### 3.2.3. Transportation, Automobile Dependence, Energy Consumption, and GHG/CO$_2$ Emissions

The transportation sector consumed more than half of the oil used globally in 2015 [59] and has been identified as the largest emitter of CO$_2$, outpacing other sectors [19]. Energy consumption and GHG/CO$_2$ emissions related to transportation, and road transportation, in particular, have seen sharp increases worldwide [6,12,55,60]. Given the fact that the vast majority of vehicles are powered by combustion engines using fossil energy, the road transport sector contributes greatly to climate change through GHG/CO$_2$ emissions.

Sprawl would have been impossible at its current scale without heavy reliance on the automobile. The general consensus in the literature is that sprawled urban forms generate more car travel and entails greater energy consumption and more GHG/CO$_2$ emissions as a consequence [4,6,19]. The research reaches the same conclusions when such outcomes are measured at the local, or neighborhood scale, or the regional level. In other words, low-density suburban environments generate more emissions than compact environments in the same city, and the more a city is marked by sprawl overall, the more emissions it generates.

**Table 5.** The impacts of land use and land cover on fossil energy consumption and GHG/$CO_2$ emissions [36] (p. 47–48).

| Author(s) Year of Publication | Scope and Location | Main Method(s) | Data/Time Frame | Land Use Factors | Main Findings |
|---|---|---|---|---|---|
| Bart (2010) | EU Member States | A simple linear multiple regression analysis | CORINE database between 1990 and 2000 | Increase in artificial land area | Sprawling development is strongly associated with increases in transport-related emissions and is the most important driver of emission growth. |
| Stone, Hess, and Frumkin (2010) | Metropolitan regions in the U.S. | Applying a widely used sprawl index | Urban form in 2000. Extreme Heat Events between 1956 and 2005 | Sprawl index, frequency of EHEs | "The rate of increase in the annual number of EHEs in the most sprawling metropolitan regions is more than twice the rate of increase observed in the most compact metropolitan regions" (p. 1425). |
| Bereitschaft and Debbage (2013) | 86 U.S. metropolitan areas | A series of linear regression models have been applied | Air pollutants data collected based on 2000 census | 5 pre-existing urban sprawl indexes were selected | After controlling other variables, higher levels of urban sprawl or sprawled urban form are closely linked with a higher level of air pollution and $CO_2$ emissions. |
| Kim, Lee, and Choi (2014) | Los Angeles Metropolitan Area (LAMA) vs. Seoul Metropolitan Area (SMA) | Comparative approach by employing the Cobb-Douglas functions | Data were collected based on the status quo from 2008 | Distinctive land-use density: an auto-centric area vs. dense, intensive land-use area | Reduction of $CO_2$ emissions in both areas can be achieved by the public transit mode share adjustment without weakening existing mobility levels. However, the amount of $CO_2$ reduction of the SMA is much more significant than that of the LAMA. |

**Table 5.** *Cont.*

| Author(s) Year of Publication | Scope and Location | Main Method(s) | Data/Time Frame | Land Use Factors | Main Findings |
|---|---|---|---|---|---|
| Adeyemi et al. (2015) | Tshwane metropolis, Gauteng Province, South Africa | a correlation analysis to test the relationship between Land Surface Temperature, Normalized Difference Vegetation Index, and Normalized Difference Built-up Index | Landsat 8 LCDM, 2003, and Landsat 7 ETM+, 2013 | Vegetation cover and impervious surface area | LST has a positive relationship with NDBI, while has a negative relationship with NDVI. |
| Wang, Li, and Yang (2015) | Southern China | A structural equation model | 1988 and 2005 | Vegetation, urban and surrounding area, and other | "Adding vegetation area is the main method to mitigate regional climate change" (p. 1). |
| Iwata and Managi (2016) | Japanese cities (1750) | Linear model | City-level data from 1990 to 2007 | Impacts of different land-use strategies | Different urban planning instruments impact the level of vehicular $CO_2$ emissions differently. Some methods are more effective in low-density cities, while others work better in high-density cities. |
| Emadodin, Taravat, and Rajaei (2016) | Tehran, Iran | MLP neutral network has been used; more detailed presentation sees p. 233. | Satellite images: every 5 years from 1975 to 2015; Local climatic data: 1990 to 2010. | IDM has been used to measure changes in aridity between 1990–2000 and 2001–2010. | Between these two time periods, the average temperature has increased from 17.43 to 18.31. More arid area has experienced greater temperature increase. |
| Lu and Liu (2016) | 287 Chinese cities: four provincial-level cities and 283 prefecture-level cities | A geographically weighted regression (GWR) model | NO2 data from 2008; SO2 data from 2007 | Urban form indexes: the compact ration index, the fractal dimension index, and the Boyce–Clark shape index | Urban form characteristics significantly affect urban air quality in China. |

**Table 5.** *Cont.*

| Author(s) Year of Publication | Scope and Location | Main Method(s) | Data/Time Frame | Land Use Factors | Main Findings |
|---|---|---|---|---|---|
| Cai et al. (2017) | Chinese and American cities | Compare and quantify the correlation among nighttime light intensity, surface thermal changes, and city size | MODIS LST and DMSP/OLS Nighttime light data sets 2001–2012 | Spatiotemporal changes of the urbanization process | In general, despite the spatial heterogeneities, light intensity increases with increasing city size. |
| Moradi and Tamer (2017) | Bursa City | Paired Samples t-Test; Holdren Model | 1984 to 2014 | The growth of the urban settlement, the growth of urban population Emissions decrease as density increases while moving towards city centers. | During 1995 to 2003, urban growth was ascribed to 65% of urban sprawl, accompanied by a loss of forests and agricultural land, and an increase of 1.36 °C monthly minimums temperature (p. 26). |

As was mentioned before, low densities and other land use characteristics of sprawl impact the transportation negative outputs in such contexts. But studies focused specifically on transportation dynamics point also to modal share and trip characteristics in sprawled versus more compact environments. Relative to more compact environments, sprawl fosters higher rates of automobile ownership. It is less amenable to public transit deployment and use. It is associated with lower levels transit ridership, is generating longer commuting times, and is increasing VMT/VKT [61,62]. Such conditions are associated with a number of other externalities, among which is the amount of space dedicated to car infrastructure itself, in the form of roads, highways, parking lots, etc., all impervious surfaces that alter the water cycle and contribute to higher ground-level temperatures.

### 3.3. On Some Gaps, Limitations, and Ambiguities in the Literature

While the consensus is strong about the environmental costs of sprawl and its contribution to climate change, there exists a number of gaps, limitations, and seemingly diverging interpretations on a limited number of specific aspects. Differences in the conceptualization of sprawl, and consequently in the measurements of its various dimensions and overall configurational patterns, affect the ability to analyze more precisely the impacts of specific urban form and land use attributes on transportation patterns and their associated GHG/$CO_2$ emissions. In addition, as already mentioned, the measurement of sprawl or some of its attributes, such as low densities, are particularly sensitive to the modifiable areal unit problem, or MAUP, according to which seemingly discordant results are due to the spatial partitioning used, or on the spatial resolution at which the analysis is conducted [40,41].

Another issue stems from entangled factors and conditions in sprawled environments (or their polar opposite the compact city). Higher urban density for instance is generally accompanied by mixed land uses and better public transit, so that variables measuring those aspects tend to correlate with one another [55], and with fossil energy use and GHG/$CO_2$ emissions. However, it is unlikely that there exists a simple relationship between urban form and travel behavior for instance. A majority of studies surveyed herein conclude that urban form exerts a significant influence on people's travel behavior mainly through influencing vehicle VMT/VKT, modal choice (public transit versus car), and modal split (between automobile, public and active transportation modes), but it is far less clear how urban design and specific land use characteristics influence people's travel [63,64]. The latter uncertainties do not invalidate or weaken the general conclusions that sprawl is associated with greater emissions levels, but they limit considerably the ability of interested parties to intervene efficiently to retrofit existing environments to reduce their environmental footprint as it is impossible to alter all aspects at once.

### 4. Discussion

Any serious attempts to measure the impacts of urbanized habitats on the environment, or to intervene on such issues with the aims of reducing their environmental footprints or to build-up resilience, require a deep understanding of the urban material and spatial forms that are manifested in the contemporary city. Proper theorization and characterization of the urban built environments constitute an essential facet of any such research effort. The conceptual ambiguities pertaining to the notion of sprawl, or the lack of unified normative or operational definitions have hindered researchers' abilities to engage with the multidimensionality of sprawl and to analyze more accurately its environmental costs.

Firstly, lacking proper theorization of urban form often leads to a crudely approximative characterization and quantification of spatial conditions (e.g., the widespread use of density indicators that does not account for the variability of spatial composition and configuration in which the same density can be manifested). Many studies compare people's travel behavior between different types of neighborhood. In most cases, a dichotomous classification is employed to compare and contrast internally homogeneous neighborhoods that are either "traditional" i.e., compact, or suburban, i.e., sprawled. There are several

problems with this categorization [65]. Bagley, Mokhtarian, and Kitamura criticized that a "binary designation of a residential neighborhood as either traditional or suburban is a distortion of reality, since some locations may have some characteristics of both types" [66] (p. 689) or present intermediary conditions on some or all aspects.

Secondly, sprawl should be and can only be fully understood in relative terms spatially, geographically, culturally, and temporarily. Sprawl can assume very different meanings in different geographical and cultural contexts, or in different urbanization development phases. Cities and regions sprawl differently by presenting varying patterns, rates, extents, and trends of sprawl. Local realities should be carefully taken into account when referring to thresholds used in other contexts. Cities are constantly being transformed and rebuilt upon themselves, often entailing densification and reshuffling of land utilization. Any stage displays conditions that are the temporary results of ongoing processes. As a consequence, sprawl must be conceived in both their space and time contexts.

Thirdly, the terms "sprawl" and "compact city" have been used to represent polar opposites situated at the ends of a spectrum. This raises several complex theoretical and methodological questions and poses significant challenges for the operationalization of these concepts, both for analytical and applied purposes. Morphologically speaking, such a dichotomic representation is fallacious. A city's spatial expansion over time produces a variety of urban configurations and combinatory patterns. Those various parts coexist in the same city. They are connected with one another and to the city as a whole. When a city expands, the properties in the new areas alter the spatial conditions and transportation dynamics of the whole city.

## 5. Conclusions

This review has shown that there is a significant amount of literature analyzing the relationships between sprawl and climate change and that there is a renewed interest in the topic [8]. These research efforts address different aspects of those relationships by pointing to a variety of direct and indirect links, depending on the factors and combination of factors considered. The fragmentation of the research landscape can challenge one's ability to build a synthetic picture. Highlighting three main streams of research contributes to bringing some clarity. Those coincide roughly with the three main characteristics of sprawl highlighted in the literature centered on that phenomenon. Yet, as seen, there remains significant explicit and implicit overlapping between the streams. This is partly due to ambiguities in the conceptualization of sprawl itself, as discussed in this paper, as well as the difficulty to untangle its defining characteristics when probing it quantitatively, or by extension, measuring its impacts. The latter considerations produce some "noise" and seemingly incongruent results, as variables used to measure land use, including density, urban form properties, and transportation tend to correlate with one another in "pure" sprawled, or compact, contexts, but express more elusive interrelations in less archetypical configurations, or when measured against different spatial partitioning (scale and spatial boundaries). In spite of such limitations, much of the evidence gathered in this exercise demonstrates that sprawl is associated with indisputable environmental costs, including climate change. What is at stake is not determining whether sprawl is sustainable, it is not, nor what factors make it suboptimal, those are known, but rather what combinations of factors are more potent relative to the outcomes. There is a growing consensus on the need to develop combined strategies that consider transportation, land-use, and urban form synergies [6,67,68]. Based on the literature reviewed, this is the way to go to cope with the inherent complexities of sprawl. Yet, the operationalization of such approaches remains highly challenging. For, research has been fragmented. It has yet to produce a cohesive framework clarifying what could be realistically expected from retrofitting urban form, land-use, and transportation systems, considered separately and, more importantly, in combination, to foster modal shifts toward public transportation and the reclaiming of the space lost to automobility.

**Funding:** This research was funded by Ouranos via the Plan d'action 2013-2020 sur les changements climatiques du Gouvernement du Québec.

**Conflicts of Interest:** The authors declare no conflict of interest.

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
