# Peer review of "Untangling Urban Sprawl and Climate Change: A Review of the Literature on Physical Planning and Transportation Drivers"

_atmosphere, doi:10.3390/atmos12050547_

Round 1

Reviewer 1 Report

I think that the approach of the manuscript is a very good attempt to bring some light to the relationship between urban sprawl and climate change, which is a key topic in the mitigation of global warming. The reduction of carbon due to transportation is one aspect to deal with, together with more rational use of land in the future, prioritizing urban regeneration and compact models for cities, rather than new urban expansions. In general terms, there is a consensus about that relation, but it is true that it still remains abstract and the definitions and numbers are not totally clear.

I agree with the authors about the difficulty in finding a synthetic picture due to the complexity of urban sprawl concept itself and the quantity of factors implied. In my view, to try to standardized or universalize measures is not possible considering such a complex topic and, at the end, there is a necessary adaptation to the specificities of the particular context. However, the revision of the state of the art is a very good exercise in order to establish a valuable starting point. In this sense, I find the manuscript very appropriate.

The text is well organized, summarized and nicely written. The tables go directly to the point and make a good summary of the points that the manuscript is trying to make. I specially find very much clarifying the Figure 1 in order to have a general picture on the matter.

Still, I find that some improvements could be done:

  1. The authors limit the references search to the period 1989-2018. The information of the tables is mainly based on previous work, reference nr 34. In order to add new content, my advice is to include some updated references. Some examples, (if the authors find that they meet the criteria of the bibliography search) could be:
  • Rubiera-Morollon, F.,Garrido-Yserte, R. (2020) Recent Literature about Urban Sprawl: A Renewed Relevance of the Phenomenon from the Perspective of Environmental Sustainability. SUSTAINABILITY, 12 (16), nr 6551.
  • Hemin Mohammed, I. Urban form study: the sprawling city-review of methods of studying urban sprawl. GEOJOURNAL. DOI: 1007/s10708-020-10157-9
  • , etc.
  1. I think that some limitations of the work should be included:
    1. The manuscript is based in transportation which is a very relevant sector. However, I think that it is important to highlight the role of buildings, considering the model based on detached houses scattered in the territory in urban sprawl model, that entails much higher energy consumption and carbon emissions, than in the compact city model.
    2. It should be also warned the convenience of promoting and increase the use of public transportation in order to reduce the use of private vehicles. I think that it is something to improve in many cities in the American context.
    3. On line work could be also a strategy to reduce transportation connected with the urban sprawl. The potential of this has been demonstrated during the Covid-19 pandemic.
    4. Therefore, it is important to point out that there should be combined strategies addressed to cover all the sectors involved.
    5. Citizen’s awareness and sensibilization is another strategic front.
    6. Minor spelling mistakes:
  • Page 1, line 13, published instead of published
  • Page 3, line 113, transport instead of [t]ransport
  • Page 4, line 143, observers instead of observer?
  • Page 13, line 395, binary instead of [b]inary

Reviewer 2 Report

The work submitted for review is an overview showing major trends in literature regarding relationships between urban sprawl and climate change since 1979. The work makes an analysis of the applied approaches to defining the investigated phenomenon and main indexes used to measure it. The content of the discussed literature is clearly arranged. The Authors point to significant gaps in the discussed research topic and areas which require further work and development. Overview works come to be published relatively seldom. However, they constitute a very important source for researchers, facilitate discussion of research results and, due to indicating existing gaps in research, may serve as inspiration to develop particular research directions. I recommend the work for publication in the journal “Atmosphere” in its present form. 

Author Response

The reviewer 2 recommended "the work for publication in the journal "Atmosphere" in its present form". The authors greatly appreciate the nice comments received from reviewer 2. The authors would like to sincerely thank the reviewer 2 for the generous comments and valuable thought. Those were most useful.